# Functional Frailty, Dietary Intake, and Risk of Malnutrition. Are Nutrients Involved in Muscle Synthesis the Key for Frailty Prevention?

**DOI:** 10.3390/nu13041231

**Published:** 2021-04-08

**Authors:** Ana Moradell, Ángel Iván Fernández-García, David Navarrete-Villanueva, Lucía Sagarra-Romero, Eva Gesteiro, Jorge Pérez-Gómez, Irene Rodríguez-Gómez, Ignacio Ara, Jose A. Casajús, Germán Vicente-Rodríguez, Alba Gómez-Cabello

**Affiliations:** 1GENUD (Growth, Exercise, NUtrition and Development) Research Group, Universidad de Zaragoza, 50009 Zaragoza, Spain; amoradell@unizar.es (A.M.); angelivanfg@unizar.es (Á.I.F.-G.); dnavarrete@unizar.es (D.N.-V.); joseant@unizar.es (J.A.C.); gervicen@unizar.es (G.V.-R.); 2Agrifood Research and Technology Centre of Aragón -IA2-, CITA-Universidad de Zaragoza, 50009 Zaragoza, Spain; 3Exercise and Health in Special Population Spanish Research Net (EXERNET), 50009 Zaragoza, Spain; eva.gesteiro@upm.es; 4Faculty of Health and Sport Science (FCSD), Department of Physiatry and Nursing, University of Zaragoza, Ronda Misericordia 5, 22001 Huesca, Spain; 5Department of Physiatry and Nursing, Faculty of Health, University of Zaragoza, 50009 Zaragoza, Spain; 6Faculty of Health Sciences, San Jorge University, Villanueva de Gállego, 50830 Zaragoza, Spain; lsagarra@usj.es; 7ImFINE Research Group, Department of Health and Human Performance, Faculty of Physical Activity and Sport Sciences-INEF, Polytechnic University of Madrid, 28040 Madrid, Spain; 8HEME (Health, Economy, Motricity and Education) Research Group, Faculty of Sport Science, University of Extremadura, 10003 Cáceres, Spain; jorgepg100@unex.es; 9GENUD Toledo Research Group, University of Castilla-La Mancha, 45071 Toledo, Spain; irene.rodriguez@uclm.es (I.R.-G.); ignacio.ara@uclm.es (I.A.); 10Biomedical Research Networking Center on Frailty and Healthy Aging (CIBERFES), 28029 Madrid, Spain; 11Centro de Investigación Biomédica en Red de Fisiopatología de la Obesidad y Nutrición (CIBERObn), 28029 Madrid, Spain; 12Centro Universitario de la Defensa, 50090 Zaragoza, Spain

**Keywords:** performance, frailty, nutritional status, vitamin D, protein

## Abstract

Frailty is a reversible condition, which is strongly related to physical function and nutritional status. Different scales are used to screened older adults and their risk of being frail, however, Short Physical Performance Battery (SPPB) may be more adequate than others to measure physical function in exercise interventions and has been less studied. Thus, the main aims of our study were: (1) to describe differences in nutritional intakes by SPPB groups (robust, pre-frail and frail); (2) to study the relationship between being at risk of malnourishment and frailty; and (3) to describe differences in nutrient intake between those at risk of malnourishment and those without risk in the no-frail individuals. One hundred one participants (80.4 ± 6.0 year old) were included in this cross-sectional study. A validated semi-quantitative food frequency questionnaire was used to determine food intake and Mini Nutritional Assessment to determine malnutrition. Results revealed differences for the intake of carbohydrates, *n*-3 fatty acids (n3), and saturated fatty acids for frail, pre-frail, and robust individuals and differences in vitamin D intake between frail and robust (all *p* < 0.05). Those at risk of malnutrition were approximately 8 times more likely to be frail than those with no risk. Significant differences in nutrient intake were found between those at risk of malnourishment and those without risk, specifically in: protein, PUFA *n*-3, retinol, ascorbic acid, niacin equivalents, folic acid, magnesium, and potassium, respectively. Moreover, differences in alcohol were also observed showing higher intake for those at risk of malnourishment (all *p* < 0.05). In conclusion, nutrients related to muscle metabolism showed to have different intakes across SPPB physical function groups. The intake of these specific nutrients related with risk of malnourishment need to be promoted in order to prevent frailty.

## 1. Introduction

Frailty is characterized by a loss of strength, endurance, and physical ability and cognitive function, which results in an increased risk of vulnerability to disease, dependence, and death [1,2]. Previously to this state and subsequently to a physiological decline, a pre-frail stage identifies a subset of high risk and potentially reversible condition before onset of established frailty [2]. Evidence shows that those in an intermediate stage between robust and frailty, namely, pre-frail, present an increased risk of becoming frail within just 3 years [3].

The most common method to assess frailty and pre-frailty stages is the Fried Phenotype [3]; however, the Short Physical Performance Battery (SPPB) is also frequently used [4] as a screening tool. Although the Fried phenotype has been widely used as a frailty scale, its applicability in the routine clinical practice is questionable because of the complexity of some measurements such as a handgrip dynamometer. Pre-disability condition of frailty can indeed be captured using the SPPB as a comprehensive measure of physical functioning impairment [5]. Even this first scale is the most widely cited [6], both have been widely used across the literature. However, components of the assessments differ, which may have implications on the feasibility of incorporating these assessments into clinical practice. For example, Pritchart et al. found different results when both scales were used to determine pre-frail and frail stages [7], and Lim et al. have recently suggested not overlapping the scales [8]. Nevertheless, the use of SPPB is increasing as it evaluates not only physical function but also physical performance through a mobility domain that could be of a higher interest in rehabilitation, physical exercise, and physical activity-related interventions.

Variety of health conditions contribute to the development of frailty, including environmental factors such as physical activity [9] or poor nutrition [10,11]. In addition, diseases as sarcopenia, defined as a generalized skeletal muscle mass disorder, often overlap with frailty and led to this detriment [12]. However, it implies not only muscle in limbs but also those involve in chewing and swallowing [13], which affects negatively food consumption. Optimal nutritional intake could delay frailty by avoiding chronic diseases such as cardiovascular diseases, obesity, and diabetes [14], increasing muscle mass and physical function, and even improving immune system [15]. In this regard, multiple studies have associated frailty, assessed by Fried phenotype, with different nutritional parameters such as low energy and low protein intake, vitamin B12 and vitamin D deficiency, or a higher risk of malnourishment [10,16]. However, to the best of our knowledge, a few studies differ between nutritional intake in these three-frailty stages determined by Fried [17,18,19], while no studies have been published reporting neither these dietary intake differences in frailty stages assessed by SPPB. This relationship could be interesting to design and implement more accurate strategies involving exercise and nutritional supplementation. There are existing exercise interventions including older adults and considering frail Fried phenotypes, which combine nutritional supplements and exercise in order to improve functional capacity [15,16,20]; they considered each nutrient’s attributed effect without considering differences in dietary intakes between SPPB stages and taking these results into account could improve their outcomes. Thus, it should be of high importance to investigate which nutrients differ in frailty stages in order to design appropriate interventions according to the stage of frailty.

In light of the above, authors from the present study hypothesize to find deficiencies and lower intakes in nutrients that are strongly related to skeletal muscle synthesis in those people classified as frail or even in pre-frail compared with robust (according to SPPB). We also hypothesize that being at risk of malnutrition increases the likelihood of being frail.

Thus main purposes of this study were: (1) to investigate the differences in dietary intake between the different stages measured by SPPB (robust, frail, and pre-frail); (2) to describe the proportion of frail, pre-frail, and robust who meet the EFSA dietary references values in older adults; (3) to study the relationship between the risk of malnourishment and the development of functional frailty (measured by SPPB); and (4) to assesses possible key nutrients associated to possible development of frailty when there is a risk of malnutrition.

## 2. Materials and Methods

### 2.1. Study Design and Participants

This cross-sectional study was carried out in the framework of the EXERNET Elder 3.0 project. Participant recruitment was done in three health care centers and three nursing homes from the city of Zaragoza, Spain during 2018. Briefly, this study aims to implement a 6-month multicomponent exercise program in frail and pre-frail older adults in order to improve physical function and physical performance. Data for this report correspond to the evaluation previous to the intervention phase. This report also includes robust individuals who came to the recruitment phase. Inclusion criteria for the study were: (1) to be older than 65 years old, (2) not suffering from dementia and/or cancer, and (3) not being invalid (<4 points scored by SPPB). Participants with missing information of the food frequency questionnaire or mini nutritional assessment score were also excluded for the present study.

All methodology was described carefully elsewhere [21]. Information about functional capacity and other health/lifestyle outcomes such as daily walking and sitting hours, smoking, cognitive status (measured by Mini Mental State Examination [22]), or sleeping hours were collected through a structured questionnaire. The dietary information (food frequency questionnaire) was obtained once, in a separated day [23].

### 2.2. Ethics Statement

Oral and written information and possible benefits and risks derived from participation in this study were given to participants during the first day of attendance. Afterwards, from all the included participants, a written informed consent was obtained. National and European legislation related to data protection was followed rigorously.

The study was performed according to the Helsinki Declaration of 1961 revised in Fortaleza (2013) and the current legislation of human clinical research of Spain (Law 14/2007). The Hospital Universitario Fundación de Alcorcón (16/50) approved the study protocol.

### 2.3. Short Physical Performance Battery

The Short Physical Performance Battery (SPPB) was performed in order to evaluate the physical performance and functional status of the participants. Three tests composed the SPPB; balance (to stand up for 10 s with feet positioned in three ways: together side-by-side, semi-tandem, and tandem positions), usual gait speed (time to complete 4 m walking), and lower limb strength (time to rise 5 times from a chair) [24].

The total battery score from 0 to 12 points. Four functional stages were created in order to classify participants: dependent (<4 points), frail (4–6 points), pre-frail (7–9 points), and robust (>9 points) [4].

### 2.4. Anthropometrics and Body Composition Measurements

A portable stadiometer of 2.10 m (SECA, Hamburg, Germany) was used to measure height. To measure body weight (kg) and to estimate body total fat mass (TFM), percentage of body fat (FM%), and fat free mass (FFM), a portable bioelectrical impedance analyzer (BIA) (TANITA BC 418-MA Tanita Corp., Tokyo, Japan) was used. To standardize and avoid bias in the process, all participants had to come to the research center early in the morning with fasting. They were also advised to empty their bladder before the measurements. Older individuals had to remove shoes and heavy clothes. Body mass index (BMI) was calculated dividing weight (in kg) by the height in meters squared ((BMI = weight/height^2^; kg/m^2^).

Mid-arm (relaxed) and calf circumferences were evaluated according to the International Society for the Advancement of Kinanthropometry (ISAK) protocol. A Rosscraft Anthrotape (Rosscraft Innovations Inc, Vancouver, BC, Canada) was used for this purpose.

### 2.5. Mediterranean Diet Adherence

The 14-point Mediterranean Diet Adherence questionnaire consist of 12 questions about food intake and 2 questions about food habits considered as characteristic from the Mediterranean diet. The result allows knowing the adherence to this diet [25]. Maximal score possibly obtain is 14, as each item point 0 or 1 depending on if the habit asked is complying each item. Results from the Mediterranean Diet Adherence were categorized as low adherence (<9 points) and high adherence (≥9 points).

### 2.6. Mini Nutritional Assessment

The Mini Nutritional Assessment (MNA) consists of 18-items 15 questions about diet, self-perception of nutritional and health state, and functional or independence and three anthropometric parameters (BMI, calf circumference and mid-arm circumference). All the items were specific for geriatric assessment. The final score classifies the participant as; well-nourished (>23.5 points), at risk of malnourishment (17–23.5 points), or malnourished (<17 points) [26,27].

### 2.7. Food Frequency Questionnaire

To assess dietary intake, a semiquantitative food frequency questionnaire, previously validated in Spain was used [23,28]. Information collected was relative to the last year. Moreover, 137 items accompanied by their typical portion size was complete. Participants selected the frequency of consumption between nine options ranging from never/almost never to six or more times per day. To obtain the daily intake, the portion size was multiplied by the frequency of consumption. Spanish food composition tables and other sources of information [29,30] were used to estimated nutrient intake. Data extracted from this questionnaire were total mean energy intake (kcal/day), macronutrients (protein, fat, and carbohydrates in g/day and % kcal of the total macronutrient energy distribution), alcohol (g and % kcal of the total macronutrient energy distribution), types of fatty acids (g/day), types of polyunsaturated fatty acids (PUFA) (*n*-3 and *n*-6) (g/day), vitamins and minerals (mg or ug/day as corresponding to the nutrient referred). Moreover, for each food item, we estimated the average amount of food consumed in grams and grouped them according to their nutrient contribution.

Dietary Reference Values were used in the study according to EFSA recommendations (2017) for adults [31].

### 2.8. Statistical Analysis

Calculations were performed using The Statistical Package for the Social Sciences (SPSS) v. 20.0 for Windows (SPSS, Inc., Chicago, IL, USA). Normality of the data was ensuring for the variables in three SPPB groups (robust, pre-frail, and frail). Differences between descriptive characteristics were assessed by analysis of variance (ANOVA) of one factor for continuous variables and chi-squared test for categorical variables. An additional ANOVA analysis was performed to describe differences between food group consumption. Groups according to dietary recommendations were created in order to describe how much people meet recommendations in each group and to show differences between SPPB groups by a chi-squared test. Moreover, an analysis of covariance (ANCOVA) adjusted by energy intake was performed to study differences in nutrient intake between the three groups. Further analyses were used to show differences in frail and pre-frail compared with robust as the reference group.

Additionally, a binary logistic regression analysis was used to study if being at risk of malnutrition was a predictor of being frailty. For this analysis, a non-frailty group was created including robust and pre-frail participants together and separated from frail, as a different group to compare them. The reason for this grouping was to increase the number of subjects and thus, the power of the analyses when comparing against frailty. Finally, differences in nutrient intake between those at risk of malnutrition and those with no risk in non-frail group were investigated by another ANOVA in order to elucidate possible key nutrients, which could influence frailty development in those no-frail, between those at risk of malnutrition, and those without risk. Statistical significance for all the analyses was set at *p* < 0.05.

## 3. Results

A total of 101 participants (78 females) with a mean age of 80.4 years met the inclusion criteria and were included in this report. Descriptive characteristics and differences between robust, pre-frail, and frail participants are shown in Table 1. Statistical differences between groups were observed for age, weight, and MNA.

Table 2 describes the intake of food group’s consumption for each SPPB group. Differences were only observed for cheese between pre-frail and robust (*p* < 0.05).

Percentages of the sample covering the Spanish DRI for vitamins and minerals in each SPPB group are shown in Table 3.

Differences in percentage of people who cover these recommendations were found for vitamin D and B12. Concretely, for vitamin D, a high proportion of robust people met the recommendations (15.4%), while anyone in the other group reach the reference values. For B12, the whole sample of robust met the recommendations (100.0%), followed by pre-frail (95,6%) and frail (70%).

Differences between the amount of nutrients consumed, adjusted by energy intake are presented in Table 4.

Globally, differences were observed between groups for carbohydrates, n3 fatty acids, and saturated fatty acids (SFA) (all *p* < 0.05). Specifically, differences were observed between robust and frail for carbohydrates (234.3 ± 12.8 vs. 279.1 ± 10.3 g/day), protein (112.6 ± 4.6 vs. 99.2 ± 3.7 g/day), n3 (3.2 ± 0.3 vs. 2.0 ± 0.2 g/day), SFA (36.1 ± 2.0 vs. 29.6 ± 1.6 g/day), and vitamin D (7.8 ± 1.0 vs. 4.6 ± 0.8 μg/day) (all *p* <0.05). In addition, differences were observed between pre-frail and frail, specifically for protein (101.6 ± 2.0 vs. 99.2 ± 3.7 g/day) and SFA (30.0 ± 0.9 vs. 29.6 ± 1.6 g/day) (both *p* < 0.05). No statistically significant differences were found between robust and pre-frail for any nutrient.

Moreover, Figure 1, Figure 2, Figure 3 and Figure 4 show differences in dietary intakes between groups when referenced versus the robust group. Data about graphs is describe detailed in a Appendix A. Figure 1 shows differences in macronutrient intake; frail presented a higher intake of carbohydrates (β = 44.9 ± 16.41) when compared to robust and a lower consumption of protein (β = −13.4 ± 6.0) and total fat (β = −13.0 ± 6.4), while pre-frail group only showed differences in protein (β = −11.0 ± 5.1) when compared to robust (all *p* < 0.05).

Data illustrating fat-type consumption are presented in Figure 2.

A lower consumption of SFA was observed in frail and pre-frail (β = −6.6 ± 2.6 and β = −6.2 ± 2.2, respectively) when considering robust as reference, and only frail presented a smaller intake of PUFA *n*-3 (β = −1.1 ± 0.4) (all *p* < 0.05).

Regarding vitamins (Figure 3), frail group showed smaller intake for niacin equivalents (β = −7.4 ± 3.2) and for pyridoxin (β = −0.4 ± 0.2), while pre-frail only showed smaller intake for niacin equivalents (β = −5.5 ± 2.7) when robust was taken as the reference (all *p* < 0.05; Figure 3).

For minerals (Figure 4), only pre-frail showed smaller intakes of phosphorus (β = −265.5 ± 118.7) when compared with the reference group of robust (*p* < 0.05).

In addition, the binary logistic regression analysis was performed to predict the probability of being frail when there is a risk of malnourishment (adjusted by sex and age) and showed that those who were at risk of malnourishment were approximately 8 times more likely of being frail in comparison to those who are not at risk of malnourishment (β = 7.7; *p* < 0.05).

Finally, differences in nutritional intake of non-frail comparing those at risk of malnutrition and those without risk are presented in Table 5.

Those non-frail participants at risk of malnourishment showed significant lower intake for protein (15.9 ± 3.3 vs. 17.6 ± 2.5% from the total energy intake), PUFA *n*-3 (2.3 ± 1.0 vs. 3.2 ± 1.7 g/day), retinol (1386.1 ± 552.8 vs. 1751.1 ± 722.4 μg/day), ascorbic acid (238.2 ± 107.9 vs. 297.3 ± 128.4 mg/day), niacin equivalents (40.4 ± 10.5 vs. 45.6 ± 10.5 mg/day), folic acid (403.9 ± 90.5 vs. 475.0 ± 131.3 μg/day), magnesium (392.2 ± 83.9 vs. 451.3 ± 111.6 mg/day), and potassium (4591.9 ± 829.8 vs. 5293.8 ± 1302.0 mg/day) (all *p* < 0.05). Moreover, those at risk of malnutrition also had a higher alcohol intake compared to the well-nourished (2.6 ± 3.6 vs. 1.1 ± 1.9 g, *p* < 0.05).

## 4. Discussion

The main findings of this study are: (1) some differences exist in the nutritional intake (carbohydrates, protein, vitamin D, PUFA *n*-3, and SFA) between robust, pre-frail, and frail older people but not for food groups; (2) vitamin D recommendations were met in higher proportions in robust group, while none of the pre-frail and pre-frail participants reached recommendations; (3) those older adults at risk of malnutrition were 7.7 times more likely to being frail compared to those without risk of malnutrition, (4) differences in intakes of protein, alcohol, PUFA *n*-3, retinol equivalents, ascorbic acid, niacin, pyridoxin, folic acid, magnesium, and phosphorus were observed between those non-frails at risk of malnourishment suggesting their important role in frailty prevention.

Nutrient deficiencies, nutrient intake, and diet quality have been widely studied in frail people determined by Fried phenotype [10,32]. Nevertheless, although SPPB has emerged as a tool for the screening of frailty in recent years, no comparable studies using this instrument have been found in the literature and, consequently, results found in this report have been compared with other studies using Fried phenotype.

Regarding differences in groups created with SPPB, frail, and pre-frail older adults showed lower consumption of protein, vitamin D, and PUFA *n*-3. These nutrients related to frailty have an important role in muscle mass synthesis during aging, sarcopenia, and inflammation [33,34]. Larger sample sizes found similar results regarding protein and vitamin D, however, in contrast to our results [35,36], no differences have been observed between frail and no-frail in other Spanish populations for PUFA *n*-3 [37]. However, it should be considered the ratio of PUFA *n*-3:*n*-6 as their assimilation depends on each other. In our study, not only frail groups presented lower consumption but also these ratios seemed to be unbalanced for those groups. Moreover, other differences such as higher intakes in carbohydrates and lower in SFA in the frail group also were found. Nevertheless, higher intakes of protein could be compensated by lower intakes of carbohydrates in robust, and the higher consumption of SFA in the robust group could be related to the quality of that protein source. If protein is obtained mainly from meat, it would lead to an increased consumption of SFA.

In addition to differences in nutrient intakes between frail, pre-frail, and robust individuals, most of our population did not meet the vitamin D recommended intakes (only 15% of the robust). This result and all the initials suggested the prioritization for the role of nutrition in frailty development, specially, when physical function is measured, or it is attempting to improve it. Thus, future exercise strategies with the main aim of improve physical function should consider all these nutrients.

Muscle mass and strength reduction due to aging may lead to muscle weakness and/or an impairment in physical function as well as physical activity, which may result in the reduction in total energy expenditure and also energy requirements [38]. Collectively, those factors could lead to complicate decrease in appetite, which is strongly related with risk of malnourishment [38]. Prior studies reveal that nutritional status could be helpful to screen frailty previously to the assessment by Fried [39]. Similarly, our study reveals that those at risk of malnutrition are approximately 8 times more likely to be frail than those without risk. Nutritional status and frailty have been also associated with quality of life [40,41]. Consequently, our results highlight even more the importance of ensure an adequate nutritional intake in this population.

Additionally, some nutrients’ levels need to be remarkable for the prevention of frailty when nutritional status is considered. Differences were observed between those at risk and those without risk of being malnourished in the non-frail group. We observed higher intakes of protein, PUFA *n*-3, retinol equivalents, ascorbic acid, niacin, pyridoxin, folic acid, magnesium, and phosphorus in those without risk of malnourishment, while higher intakes of alcohol were observed in those at risk. Once more, PUFA *n*-3 and protein intake show their importance in these physiological statuses. Meanwhile, other nutrients such as vitamin A and ascorbic acid appear related to the risk of malnourishment and have been previously suggested to mediate in frailty due to their antioxidant effect, which may facilitate muscle mass synthesis [42]. Likewise, the importance of B group vitamins for their role in blood cell formation, macronutrient metabolism, and cognitive function, among others, has been also widely studied. Despite pyridoxin and vitamin B12 have been more associated in literature with frailty [43], folic acid and niacin seem also to be relevant in our sample. Furthermore, minerals such as magnesium and potassium, which appear to be significant in our sample, have been also associated with frailty and sarcopenia [44,45].

Globally, our results led us to focus not only on nutrients related to muscle mass synthesis but also on protein intake when approaching the dietary side of frailty. Moreover, future interventions using supplements, as those developed during last years with PUFA *n*-3 [46], vitamin D [47], or protein [48] and which have been demonstrated to be effectiveness to prevent frailty, should consider differently frailty stages. Interestingly, recent studies have reported different responses to protein intake between those stages [49], supporting our recommendation.

Limitations of this study should be highlighted. The present study has a cross-sectional design, reflecting associations but not revealing causality. Further research including larger sample sizes is required to verify these results in representative populations. Comparable groups with equal number of participants from both sexes need also to be performed. Although food frequency questionnaire is a validated method, this population could be over or underestimating their intakes as it is shown in the table of food group intake. Moreover, it could be also influenced by a possible cognitive impairment, however, no participants with these problems were included in this study as shown in the cognitive assessment. Some strengths like harmonized assessments and well instructed researchers should be considered as well as the novelty and practical potential of the topic. Finally, other variables such as quality of diet and nutrient food sources should be taken into account for future analyses.

## 5. Conclusions

In summary, our results showed lower intakes of protein, PUFA *n*-3, and vitamin D in frail group, while revealing higher intakes of carbohydrates. Moreover, those at risk of malnutrition have almost 8 times more probabilities to develop frailty. To prevent frailty, higher intakes in protein, PUFA *n*-3, retinol, ascorbic acid, folic acid, pyridoxin, niacin, magnesium, and potassium should be promoted in those at risk of malnutrition. Thus, it is an important role of nutritionists and dietitians to ensure healthy and specific diets in age populations and to stablish nutritional guidelines according to their functional capacity.

## Figures and Tables

**Figure 1 nutrients-13-01231-f001:**
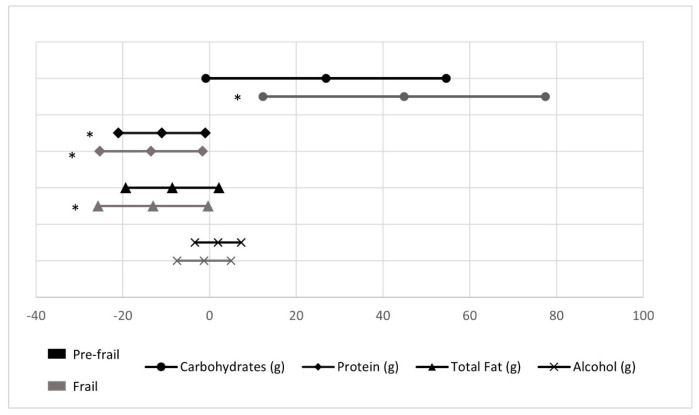
Macronutrient and alcohol intake in pre-frail and frail groups compared with robust (reference group). * Statistically significant differences (*p* value < 0.05).

**Figure 2 nutrients-13-01231-f002:**
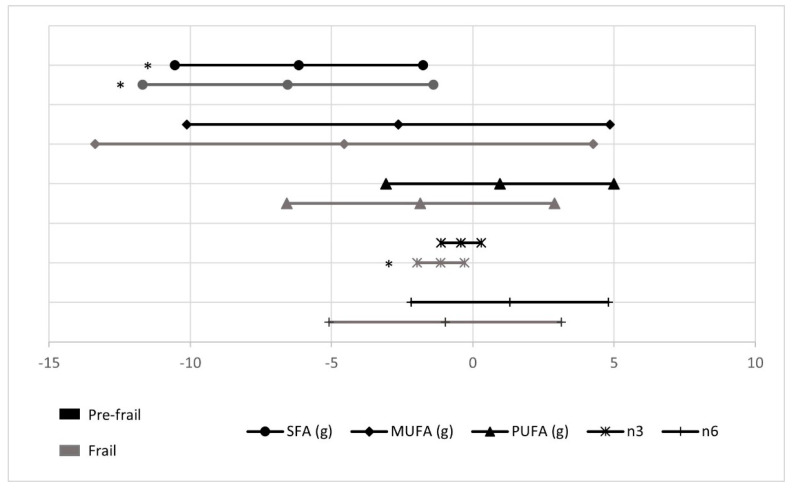
Fat type intake in pre-frail and frail groups compared with robust (reference group). * Statistically significant differences (*p* value < 0.05).

**Figure 3 nutrients-13-01231-f003:**
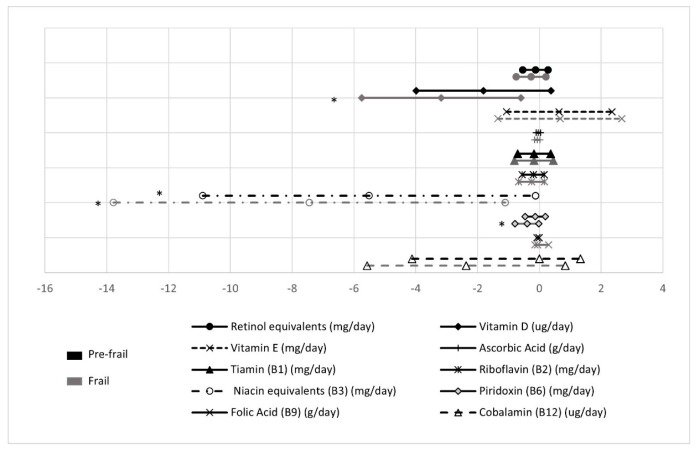
Vitamin intake in pre-frail and frail groups compared with robust (reference group). * Statistically significant differences (*p* value < 0.05).

**Figure 4 nutrients-13-01231-f004:**
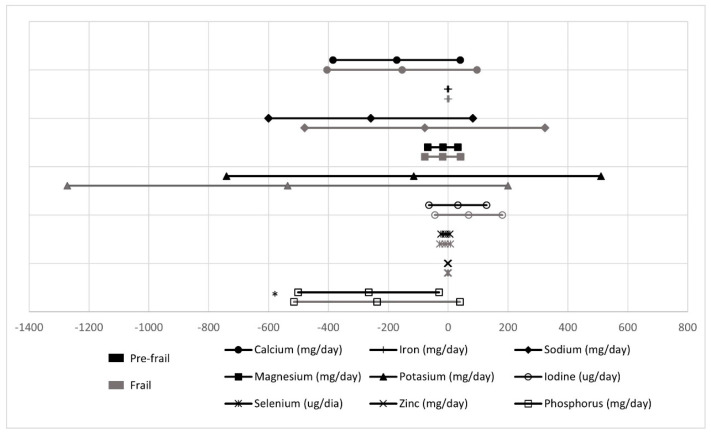
Mineral intake in pre-frail and frail groups compared with robust (reference group). * Statistically significant differences (*p* value < 0.05).

**Table 1 nutrients-13-01231-t001:** Descriptive characteristics of the participants of the study.

	Whole Sample (*n* = 101)	Robust (*n* = 13)	Pre-Frail (*n* = 68)	Frail (*n* = 20)	*p* Value
Sex					0.317
Males	23 (22.8)	1 (7.7)	18 (26.5)	4 (20.0)	
Females	78 (77.2)	12 (92.3)	50 (73.5)	16 (80.0)	
Age (years)	80.4 ± 6.0	77.3 ± 5.4	80.0 ± 5.8	83.0 ± 5.7	0.005
BMI (kg/m^2^)	29.4 ± 5.4	27.2 ± 3.0	29.9 ± 5.0	29.4 ± 7.1	0.262
Weight (kg)	72.3 ± 14.2	66.0 ± 7.0	74.8 ± 13.7	69.2 ± 16.5	0.038
BF%	37.4 ± 6.9	37.6 ± 4.9	37.9 ± 7.1	36.2 ± 7.4	0.569
FFM (kg)	44.7 ± 8.7	41.0 ± 3.5	46.2 ± 9.3	43.1 ± 8.4	0.061
MNA	23.1 ± 3.1	24.6 ± 1.9	23.6 ± 3.0	21.2 ± 2.8	<0.001
Risk of malnutrition	42 (41.6)	3 (23.0)	23 (33.8)	16 (80.0)	<0.001
No risk of malnutrition	59 (58.4)	10 (77.0)	45 (66.2)	4 (20.0)	
ADM	7.6 ± 0.95	8.3 ± 2.4	7.5 ± 1.2	7.5 ± 2.7	0.363
Low ADM	78 (77.2)	7 (53.8)	54 (79.4)	17 (85.0)	
High ADM	23 (22.8)	6 (46.2)	14 (20.6)	3 (15.0)	
Smoking	3 (3.0)	0 (0.0)	3 (4.4)	0 (.055)	0.643
MMSE	26.6 ± 2.8	27.0 ± 3.0	26.8 ± 2.6	25.5 ± 2.8	0.134

BMI: body mass index, BF%: body fat percentage, FFM: fat free mass, SPPB: short physical performance battery, MNA: mini nutritional assessment, ADM: Adherence to Mediterranean Diet, MMSE: Mini mental state examination. *n* and (%) for categorical variables, mean and standard deviation for continuous variables. All statistical significance was set in *p* < 0.05.

**Table 2 nutrients-13-01231-t002:** Differences of food group intakes between robust, pre-frail, and frail older adults.

	Robust (*n* = 13)	Pre-Frail (*n* = 68)	Frail (*n* = 20)	*p* Value
Yogurt (g/day)	71.2 ± 64.0	77.6 ± 62.0	88.2 ± 90.5	0.735
Milk (g/day)	190.0 ± 180.7	228.3 ± 167.5	300.0 ± 184.2	0.120
Cheese (g/day)	61.3 ± 41.1	27.0 ± 24.4 *	37.2 ± 44.1 *	0.002
Eggs (g/day)	27.4 ± 13.5	25.5 ± 12.8	25.1 ± 15.9	0.882
Red meat (g/day)	62.8 ± 51.0	57.7 ± 39.6	49.64 ± 6	0.614
White meat (g/day)	81.2 ± 71.7	64.6 ± 32.4	61.3 ± 41.1	0.331
Lean meat products (g/day)	32.6 ± 23	23.8 ± 20.9	19.1 ± 16.2	0.161
Fat meat products (g/day)	14.3 ± 19.5	11.4 ± 11.7	13.2 ± 12.6	0.671
White fish (g/day)	51.0 ± 30.1	45.5 ± 29.0	45.0 ± 30.5	0.809
Oily fish (g/day)	36.2 ± 41.1	25.2 ± 25.7	15.8 ± 17.5	0.079
Seafood (g/day)	25.3 ± 29.4	22.9 ± 28.6	30.4 ± 35.9	0.490
Vegetables (g/day)	463.9 ± 237.2	444.6 ± 218.1	391.34 ± 221.6	0.511
Fruit (g/day)	481.9 ± 222.0	454.4 ± 236.0	517.5 ± 637.6	0.753
Nuts (g/day)	42.1 ±43.30	33.6 ± 56.7	36.76 ± 58.7	0.871
Legumes (g/day)	22.8 ± 12.9	22.3 ± 13.5	28.6 ± 19.8	0.195
Cereals and potatoes (g/day)	169.9 ± 95.7	210 ± 106.6	200.5 ± 81.0	0.402
Olive oil (g/day)	32.9 ± 21.1	31.4 ± 21.6	31.1 ± 18.1	0.964
Fats and other oils (g/day)	4.5 ± 5.5	5.0 ± 6.7	3.86 ± 5.4	0.728
Fruit juices and beverages (g/day)	84.5 ± 91.9	67.3 ± 97.4	90.8 ± 127.3	0.964
Coffee and tea (g/day)	73.3 ± 52.1	57.6 ± 44.2	48.4 ± 58.5	0.331
Savory snacks (g/day)	48.3 ± 58.3	52.8 ± 71.1	92.3 ± 96.6	0.076
Sweet snacks (g/day)	92.1 ± 74.3	100.1 ± 75.6	107.6 ± 62.9	0.817
Alcoholic consumers	(*n* = 10)	(*n* = 54)	(*n* = 15)	
Beer (g/day)	65.7 ± 101.9	77.5 ± 169.0	13.8 ± 36.4	0.337
Wine (g/day)	48.8 ± 45.72	68.4 ± 76.8	66.6 ± 68.1	0.732

* differences between pre-frail and robust groups. *p* value stablished at <0.05. Beer and wine intake differences were calculated with alcohol consumers.

**Table 3 nutrients-13-01231-t003:** Adequate intake or population reference intake and percentage of the sample covering recommendation from EFSA (European Food Safety Authority) of vitamins and minerals by Short Physical Performance Battery (SPPB) groups.

Nutrient Intake	AI or PRI (M/F)	Robust (%)	Pre-Frail (%)	Frail (%)	*p* value
Retinol equivalents (ug/day)	**750/650**	100	97.1	95.0	0.710
Vitamin D (μg/day)	15	15.4	0.0	0.0	0.001
Vitamin E (mg/day)	13/11	53.8	47.1	40	0.730
Ascorbic acid (C) (mg/day)	**110/95**	100.0	100.0	100.0	NC
Thiamine (B1) (mg/day)	**1/0.8**	100.0	100.0	100.0	NC
Riboflavin (B2) (mg/day)	**1.6**	100.0	77.9	70.0	0.105
Niacin equivalents (B3) (mg/day)	**15.4/12.5**	100.0	100.0	100.0	NC
Pyridoxin (B6) (mg/day)	**1.7/1.6**	100.0	94,1	90.0	0.494
Folic acid (B9) (μg/day)	**330**	92.3	85.3	72.0	0.373
Cobalamin (B12) (ug/day)	4	100.0	95.6	70.0	0.001
Calcium (mg/day)	**950**	76.9	60.3	76.9	0.427
Iron (mg/day)	**11**	100.0	100.0	100.0	NC
Magnesium (mg/day)	300	100.0	91.2	90.0	0.519
Potassium (mg/day)	3500	100.0	92.6	90.0	0.528
Iodine (μg/day)	150	84.6	80.9	85.0	0.887
Selenium (μg/day)	70	92.3	83.8	92.3	0.417
Zinc (mg/day)	**16.3/12.7**	69.2	38.2	35.0	0.092
Phosphorus (mg/day)	550	100.0	100.0	100	NC

AI: adequate intake represented in ordinary type; PRI: Population Recommended Intake, presented in bold type; dietary recommended intakes; M/F: values of reference for males and females; SPPB short physical performance battery; NC: not calculated.

**Table 4 nutrients-13-01231-t004:** Differences between SPPB groups in nutrients adjusted by energy intake.

Nutrient intake	Robust (*n* = 13)	Pre-Frail (*n* = 68)	Frail (*n* = 20)	*p* Value
	Mean (SD)	Mean (SD)	Mean (SD)	
Carbohydrates (g/day)	234.3 ± 12.8 ^a^	261.1 ± 5.6	279.1 ± 10.3	0.027
Protein (g/day)	112.7 ± 4.6 ^a^	101.6 ± 2.0 ^b^	99.2 ± 3.7	0.062
Total fat (g/day)	114.9 ± 5.0	106.3 ± 2.2	101.9 ± 4.0	0.130
Alcohol (g/day)	4.1 ± 2.5	6.1 ± 1.1	2.9 ± 2.0	0.319
*n*-3 (g/day)	3.2 ± 0.3 ^a^	2.8 ± 0.1	2.0 ± 0.2	0.018
*n*-6 (g/day)	14.2 ± 1.6	15.5 ± 0.7	13.3 ± 1.3	0.277
MUFA (g/day)	51.5 ± 3.5	48.8 ± 1.5	46.9 ± 2.8	0.591
PUFA (g/day)	18.0 ± 1.9	19.0 ± 0.8	16.2 ± 1.5	0.254
SFA (g/day)	36.1 ± 2.0 ^a^	30.0 ± 0.9 ^b^	29.6 ± 1.6	0.018
Retinol equivalents (μg/day)	1737.9 ± 189.3	1612.1 ± 82.6	1470.0 ± 152.2	0.530
Vitamin D (μg/day)	7.8 ± 1.0 ^a^	6.0 ± 0.4	4.6 ± 0.8	0.054
Vitamin E (mg/day)	10.9 ± 0.8	11.5 ± 0.3	11.6 ± 0.6	0.745
Ascorbic acid (mg/day)	306.3 ± 32.7	273.0 ± 14.3	237.0 ± 26.3	0.246
Thiamine (B1) (mg/day)	2.8 ± 0.3	2.6 ± 0.1	2.6 ± 0.2	0.801
Riboflavin (B2) (mg/day)	2.4 ± 0.2	2.2 ± 0.7	2.1 ± 0.1	0.460
Niacin equivalents (B3) (mg/day)	48.5 ± 2.5	43.0 ± 1.1	41.1 ± 2.0	0.063
Pyridoxin (B6) (mg/day)	2.8 ± 0.2	2.6 ± 0.6	2.4 ± 0.1	0.076
Folic acid (B9) (μg/day)	461.1 ± 33.8	450.4 ± 14.7	405.1 ± 27.2	0.294
Cobalamin (B12) (ug/day)	11.2 ± 1.3	9.8 ± 0.6	8.9 ± 1.0	0.346
Calcium (mg/day)	1299.7 ± 98.2	1127.5 ± 42.8	1145.4 ± 78.9	0.278
Iron (mg/day)	18.1 ± 1.0	17.8 ± 0.4	17.75 ± 0.8	0.956
Sodium (mg/day)	2710.1 ± 157.6	2451.3 ± 68.7	2632.1 ± 126.7	0.204
Magnesium (mg/day)	446.3 ± 13.3	428.7 ± 10.2	427.8 ± 18.7	0.777
Potassium (mg/day)	5158.5 ± 288.8	5043.7 ± 126.0	4622.2 ± 32.2	0.227
Iodine (ug/day)	280.7 ± 44.2	312.7 ± 19.3	349.2 ± 35.6	0.467
Selenium (μg/day)	112.5 ± 6.9	103.1 ± 3.0	102.3 ± 5.6	0.431
Zinc (mg/day)	14.0 ± 0.6	13.0 ± 0.3	13.2 ± 0.5	0.389
Phosphorus (mg/day)	2152.1 ± 108.7	1886.7 ± 47.4	1914.4 ± 87.4	0.086

Omega *n*-3, *n*-6: alpha linoleic fatty acid, MUFA: monounsaturated fatty acid, PUFA: polyunsaturated fatty acid, SFA: saturated fatty acid, SPPB short physical performance battery. ^a^ statistical difference between robust and frail groups ^b^ statistical differences between frail and pre-frail groups. All statistical significance was stablished at *p* < 0.05.

**Table 5 nutrients-13-01231-t005:** Differences in nutrients intake between malnutrition groups in non-frail participants.

Nutrient Intake	At Risk of Malnutrition (*n* = 26)	No Risk of Malnutrition (*n* = 55)	*p* Value
	Mean (SD)	Mean (SD)	
Energy (kcal)	2485.3 ± 619.0	2444.5 ± 557.0	0.767
Carbohydrates (%)	42.3 ± 7.7	42.2 ± 7.2	0.959
Protein (%)	15.9 ± 3.3	17.6 ± 2.5	0.013
Total fat (%)	39.2 ± 7.0	39.1 ± 6.9	0.939
Alcohol (g)	2.6 ± 3.6	1.1 ± 1.9	0.020
*n*-3 (mg/day)	2.3 ± 1.0	3.2 ± 1.7	0.016
*n*-6 (mg/day)	15.3 ± 7.8	15.4 ± 7.7	0.924
MUFA (%)	18.4 ± 4.8	17.7 ± 4.7	0.559
PUFA (%)	6.3 ± 2.0	7.0 ± 2.7	0.258
SFA (%)	11.3 ± 2.7	11.2 ± 3.0	0.846
Retinol equivalents (ug/day)	1386.1 ± 552.8	1751.1 ± 722.4	0.025
Vitamin D (μg/day)	5.8 ± 3.3	6.5 ± 4.1	0.478
Vitamin E (mg/day)	11.1 ± 3.4	11.6 ± 3.2	0.505
Ascorbic acid (C) (mg/day)	238.2 ± 107.9	297.3 ± 128.4	0.041
Thiamine (B1) (mg/day)	2.5 ± 1.0	2.7 ± 0.9	0.290
Riboflavin (B2) (mg/day)	2.1 ± 0.6	2.7 ± 0.9	0.095
Niacin equivalents (B3) (mg/day)	40.4 ± 10.5	45.6 ± 10.5	0.040
Pyridoxin (B6) (mg/day)	2.4 ± 0.6	2.8 ± 0.6	0.003
Folic acid (B9) (μg/day)	403.9 ± 90.5	475.0 ± 131.3	0.014
Cobalamin (B12) (μg/day)	8.8 ± 3.8	10.7 ± 4.9	0.096
Calcium (mg/day)	1067.6 ± 325.5	1200.3 ± 402.8	0.146
Iron (mg/day)	16.5 ± 3.8	18.5 ± 4.6	0.055
Sodium (mg/day)	2487.9 ± 688.7	2506.5 ± 907.0	0.931
Magnesium (mg/day)	392.2 ± 83.9	451.3 ± 111.6	0.019
Potassium (mg/day)	4591.9 ± 829.8	5293.8 ± 1302.0	0.014
Iodine (μg/day)	272.2 ± 155.1	325.1 ± 161.1	0.167
Selenium (ug/day)	95.1 ± 32.9	109.4 ± 31.4	0.061
Zinc (mg/day)	12.3 ± 3.5	13.6 ± 3.3	0.090
Phosphorus (mg/day)	1795.1 ± 440.6	1997.7 ± 513.5	0.087

Omega *n*-3: alpha linolenic fatty acid, *n*-6: alpha linoleic fatty acid, MUFA: monounsaturated fatty acids, PUFA: polyunsaturated fatty acids, SFA: saturated fatty acids. %: percentage of total energy intake. Statistical significance stablished at *p* < 0.05.

## Data Availability

The data are not publicly available due to privacy.

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
