# Peer review of "Functional Frailty, Dietary Intake, and Risk of Malnutrition. Are Nutrients Involved in Muscle Synthesis the Key for Frailty Prevention?"

_nutrients, 2021, doi:10.3390/nu13041231_

Round 1
Reviewer 1 Report
It is an interesting article with a potential clinical meaning. However, there are several points that need to be addressed.
I think that the abstract is too long and a little dispersive.
Page 2; line 64-66: In the routine clinical practice, the Fried phenotype scale may be of difficult implementation since it requires de novo measurements and instruments which will be not available in most settings (i.e., handgrip dynamometer). The Fried phenotype may however be helpful for screening purposes but is frequently used (often improperly) for other purposes. Furthermore, there are a variety of operationalization of this instrument.
Page 2; line 75: Please change “Variety of health outcomes” in “Variety of conditions”
Page 3; 113-114: This part may be moved to the results section but not here.
Page 3; line 118: “The dietary record”…. FFQ or dietary record, which one?
I suggest to outline in the introduction the overlap between frailty and sarcopenia especially in the physical function domain (i.e., reduced physical performance through SPPB) adding this reference doi.org/10.3390/nu11122898.
Page 12; line 284: I think that the discussion may be improved. It seems redundant in some parts.
Page 12; line 287: I suggest to change the term elderly with older people or something else (i.e., older adults, older individuals) since the term elderly is pejorative and reductionist.
Reviewer 2 Report
Title: Functional frailty, Dietary intake and risk of Malnutrition. What could be the key nutrients to prevent frailty?
The purpose of this study was to explore the difference between malnutrition and non-malnutrition in the strong elderly or the weak elderly. This research has practical reference value. But there are the following suggestions to the author.
This study used questionnaires to evaluate the food intake, frequency and other questionnaires of the elderly. The questionnaire assessed whether the cognitive function and the cognitive ability of frail elderly affected the validity of the questionnaire.
There was only one male in the healthy group, and the difference in the number of men and women was too large. It had not a positive influence on the survey sampling throughout the study. The above factors may lead to deviations in the research results.
In Robust, pre-frail, and frail groups, the different groups tested tend to frail with increasing age. Regarding the risk of malnutrition, the aging of the elderly in this study was caused by nutrition or age, which needs to be clearly discussed in the article.
It is recommended that the author strengthen the drawing of the chart to make it exquisite and easy to read.
Table 4. “SFA (g/day, PUFA(g/day, n-6(g/day)”all have less scraping on the right sides.
Table 1. BMI (kg/m2), where "2" is superscript.
Author Response
Please see the attachement

Reviewer 3 Report
Reviewer Recommendation and Comments for Manuscript #: Nutrients-1148402:
“Functional Frailty, Dietary Intake and Risk of Malnutrition. What could be the key nutrients to prevent frailty?”
GENERAL:
The current manuscript is a very interesting and important investigation aiming to describe and reveal associations with nutritional intake, physical function, and the risk for malnutrition in a cohort of aged subjects that span the spectrum of health. These broad areas of research have been and continue to present a burden globally and necessitate deeper investigation. Overall, this manuscript requires focus, and because of the lack of specificity is difficult to follow due to excessive verbiage. The authors present multiple purposes/aims for their current study yet provide no hypothesis(es) that may be projected with respect to analyzing relationships/correlative data between factors/groups. The methods are presented adequately (though the statistical section can be improved) as well as the results, which provide evidence of associations; yet, greater detail is required for the “pooling” of group data as a single point-of-reference. The Discussion must be shortened for succinctness and focus which will aid in clarity. Additionally, numerous editorial/grammatical modifications are present and should be reworked/reworded for clarity. These concepts should be considered for a major revision/modification of the current manuscript, for the team’s current research is a very important aspect of contemporary gerontological research.
TITLE
The secondary verbiage of the title presents as an all-encompassing approach to “find” some underpinning factor(s) for the prevention of frailty. The authors should consider focusing the verbiage of the title to make it very succinct (i.e. their specific reference to nutrients involved in muscle mass and/or muscle metabolism)
ABSTRACT
Page 1, Line 26: Insert “be” following “could” and Change “delay” to “delayed”
Page 1, Line 26: Delete “its onset and progression”
Page 1, Line 27: Insert “The” prior to “Main” and Change “Main” to “main”
Page 1, Line 29: Change “people” to “individuals”
Page 1, Line 30: Insert “the” prior to “main”
Page 1, Line 30: Insert “1)” prior to “…to describe”
Page 1, Line 32: Insert “2)” prior to “…and to describe”
Page 1, Line 35: Add “Hyphens” to “semi-quantitative” and “food-frequency”
Page 1, Line 36: Change “showed” to “revealed”
Page 1, Line 37: Change “…differences in the intake for carbohydrates…” to “…differences for the intake of carbohydrates…”
Page 1, Line 37: Add “individuals” following “robust” – Both occurrences
Page 1, Line 39: Change “almost” to “approximately” and Change “like” to “likely”
Page 1, Line 39: Delete “Some” prior to “significant…”
Page 1, Line 46: Add “for” following “…intake”
Page 1, Line 46: The authors should clarify the final point of this sentence related to “…showing higher intake for those at risk…” – Risk of what?
INTRODUCTION
Page 2, Line 56: Add “Comma” between “…adults, there…”
Page 2, Line 64: Change “complex” to “complexity”
Page 2, Line 67: Add “the” between “…across the literature.”
Page 2, Line 72: Add “Comma” following reference “[7]”
Page 2, Lines 82-83: Change “nutrition” to “nutritional”
Page 2, Line 83: Change “and” to “while” and Add “Comma following “[16-18]”
Page 2, Line 86: Delete “Even” prior to “There…”
Page 2, Line 89: Change “…nutrient attributed effect…” to “…nutrient’s attributed effect…”
Page 2, Line 91: Change “…study deeply…” to “…investigate…”
Page 2, Line 94: Change “…to study the…” to “…to investigate…”
Page 2, Final Paragraph: The authors should present an explicit hypothesis(es) here, given the authors do investigate and assess quantitative data in terms of associations related to the development of frailty, nutritional intake, and physical function.
MATERIALS and METHODS
2.1 Study Design:
Page 3, Line 108: Reverse the words “…includes also…” to “…also includes…”
Page 3, Line 109: Change “…people…” to “…individuals…”
Page 3, Line 109: Insert “1)” following “…were:”
Page 3, Line 110: Insert “2” and “3)” following “…old,” and “…and”, respectively
Page 3, Line 111: Insert “Hyphen” between “food-frequency”
Page 3, Line 142: Would the authors please provide greater detail as to when the BIA measurements/analyses were conducted, given that there is known influence on the outcome metrics investigated (e.g. percentage of body fat [%FM]) depending on timing of acquisition?
Page 4, Line 178: Delete “extra space” between “Values and were”
Page 4, Line 194: Change “frailty” to “frail”
Page 4, Lines 194-196: Would the authors reword these sentences for greater clarity and succinctness? It is inferred by the reviewer that the authors collapsed, “pooled”, their groups/cohorts to increase the number of subjects (and, thus power) when comparing versus the study’s “Frail Group”, however these sentences should be rephrased to make this explicit.
Page 5, Lines 198-199: Either included in this final sentence or in an additional sentence following, the authors should provide what they set the study’s alpha-level (0.05; 0.01, etc.,) as well as describe what they accepted as significance for the study’s p-value. Even though the authors do indicate p-value(s) (e.g. p<0.05) in Table(s) and Figure Legends, this should be explicitly described in the statistical analysis paragraph of the Materials and Methods Section.
RESULTS
Page 7, Line 224: Add “Comma” between “…(15.4%) and while…”
Page 8, Line 238: Delete “some” prior to “differences”
Page 8, Line 242-243: Change “…groups considering robust as reference” to “…groups when referenced versus the robust group.” If the authors prefer to modify this sentence utilizing different verbiage, that is acceptable; however, as it is written currently it is very confusing.
Page 8, Line 244: Change “smaller” to “lower”
Page 9, Line 252: Change “smaller” to “lower”
Page 9, Lines 253-254: Change “…when considering robust as reference and only frail presented a smaller intake…” to “…when the robust group was utilized as the reference group versus the frail group and presented with a lower intake…”
Page 9, Line 255: Change “smaller” to “lower”
Page 9, Line 256: Change “smaller” to “lower”
Page, 11, Line 266: “Insert “was” following “analyses”
Page 11, Line 267: Change “frailty” to “frail” and Insert “and” following “…sex and age)”
Page 11, Line 268: Change “almost” to “approximately”
Page 12, Lines 281-283: The final two sentences of this paragraph (carried over from Page 11) should be revised/reworded for clarity. Currently, it is too difficult to interpret the authors presentation of the data as well as comparisons.
DISCUSSION
Overall, the Discussion Section needs to be shortened. The authors should focus on the main findings detailed in their results and emphasize these key points as they list them in the initial paragraph (i.e. paragraph x of the discussion, main finding/key point #1; paragraph y, main finding/key point #2, etc.,).
Page 12, Line 285: Insert “The” prior to “main”
Page 12, Lines 299-304: This paragraph should be revised/reworded for clarity; as it is written it is very confusing, especially with respect to the comment concerning “…they usually be wrongly estimated by the interview.”
Page 12, Line 307: Change “smaller” to “lower”
Page 12, Line 308: Change “bigger” to “larger” or “increased sample size”
Page 12, Line 309: Delete “all” following “meet”
Page 12, Line 314: Change “smaller” to “lower”
Page 12, Line 315: Insert “these” between “also and ratios”
Page 12, Line 319: Change “contrast” to “contrasting”
Page 12, Line 320: Reverse the words “be also” to “also be”
Page 12, Line 327: Insert “the” following “example”
Page 12-13, Lines 333-334: Change “It confers higher importance to the role…” to “Prioritization for the role…”
Page 13, Line 335: Do the authors intend to use the word “pretended”? A better or more appropriate word choice is needed or rewording the sentence to convey the thought/idea the authors are attempting to deliver is apparent.
Page 13, Line 336: Change “improve” to “improving”
Page 13, Line 338: Change “aging leads to” to “aging may lead to”
Page 13, Line 340: Change “All those factors could lead…” to “Collectively, these factors could lead…”
Page 13, Line 342: Change “previously” to “prior”
Page 13, Line 344: Change “almost” to “approximately”
Page 13, Line 348: Change “…nutrients need…” to “…nutrients’ levels need…”
Page 13, Line 355: Change “appear in relation to…” to “appear related to…”
Page 13, Line 356: Insert “Comma” between “effect, “which”
Page 13, Line 360: Insert “does” between “which does not
Page 13, Line 362: Delete “also”
Page 13, Line 365: Delete “extra space” between “pyridoxin and”
Page 13, Line 371: Delete “all”
Page 13, Lines 372-376: Delete entire sentence indicating “recommendation(s)”
Page 13, Line 379: Change “In this line,” to “Interestingly,”
Page 13, Lines 380-384: Delete “, supporting our recommendation.” And Delete the final two sentences of this paragraph for succinctness and clarity.
Page 13, Line 385: Delete “Some strengths and” and Begin this paragraph’s sentence with “Limitations of this study…”
Page 14, Lines 390-391: Delete sentence beginning with “However,…”
Page 14, Line 395: Change “To sum up…” to “In summary,…”
Page 14, Line 396: Insert “Comma” following “group” and Change “showing” to “revealing”
Page 14, Line 397: Change “almost” to “approximately”
Page 14, Lines 397-399: This sentence needs to be reworded starting with the initial phrase “To prevent frailty…”
Page 14, Line 400: Change “…seems important the role…” to “is an important role…”
Page 14, Line 401: Change “older people” to “age individuals or populations”
TABLES/FIGURES; TABLE/FIGURE LEGENDS
Page 8, Line 249: Change “…about fat type…” to “”…illustrating fat-type…”
Page 10, Figure 3 and Figure 4 Legends: The text for both figure legends is very difficult to read; please consider increasing the size of the font for readership.
REFERENCES
None
Author Response
Please see the attachement

Round 2
Reviewer 1 Report
All my comments have been addressed
Reviewer 3 Report
The authors have made all suggested and necessary revisions raised by this reviewer (and have done so in a very timely manner), which has improved the overall succinctness and clarity of their report. Their manuscript is now acceptable for consideration for publication.